

# Diverse bacterial communities exist on canine skin and are impacted by cohabitation and time

Sheila Torres[1], Jonathan B. Clayton[2], Jessica L. Danzeisen[2], Tonya Ward[3], Hu Huang[3], Dan Knights[3,4] and Timothy J. Johnson[2]

[1] Department of Veterinary Clinical Sciences, University of Minnesota, Saint Paul, MN, USA
[2] Department of Veterinary and Biomedical Sciences, University of Minnesota, Saint Paul, MN, USA
[3] Biotechnology Institute, University of Minnesota, Minneapolis, MN, USA
[4] Department of Computer Science and Engineering, University of Minnesota, Minneapolis, MN, USA

Corresponding author
Timothy J. Johnson,
joh04207@umn.edu

## ABSTRACT

It has previously been shown that domestic dogs and their household owners share bacterial populations, and that sharing of bacteria between humans is facilitated through the presence of dogs in the household. However, less is known regarding the bacterial communities of dogs, how these communities vary by location and over time, and how cohabitation of dogs themselves influences their bacterial community. Furthermore, the effects of factors such as breed, hair coat length, sex, shedding, and age on the canine skin microbiome is unknown. This study sampled the skin bacterial communities of 40 dogs belonging to 20 households longitudinally across three seasons (spring, summer, and winter). Significant differences in bacterial community structure between samples were identified when stratified by season, but not by dog sex, age, breed, hair type, or skin site. Cohabitating dogs were more likely to share bacteria of the skin than non-cohabitating dogs. Similar to human bacterial microbiomes, dogs' microbiomes were more similar to their own microbiomes over time than to microbiomes of other individuals. Dogs sampled during the same season were also more similar to each other than to dogs from different seasons, irrespective of household. However, there were very few core operational taxonomic units (OTUs) identified across all dogs sampled. Taxonomic classification revealed *Propionibacterium acnes* and *Haemophilus* sp. as key members of the dog skin bacterial community, along with *Corynebacterium* sp. and *Staphylococcus epidermidis*. This study shows that the skin bacterial community structure of dogs is highly individualized, but can be shared among dogs through cohabitation.

Subjects Microbiology, Veterinary Medicine, Dermatology
Keywords Canine, Skin, Longitudinal, Bacterial community, Co-habitation, Microbiome

## INTRODUCTION

The skin contains a very effective physical, immunological, and microbial barrier that protects the body against dehydration and constant environmental insults. The bacterial communities of the skin have been well studied, and computational and laboratory

advances in the technology of microbial community profiling have enabled more accurate investigation of these communities commonly referred to as the microbiota or microbiome. Various studies using next generation sequencing techniques have shown that the skin bacterial community of healthy humans is quite diverse and its composition, biomass, and diversity are highly influenced by the physiological characteristics of the cutaneous microenvironment (*Costello et al., 2009*; *Grice et al., 2008*, *2009*; *Grice & Segre, 2011*). Additional studies have shown that age and environmental factors such as cohabitation and having a dog also influence the composition and diversity of the skin bacterial community of healthy humans (*Capone et al., 2011*; *Dominguez-Bello et al., 2010*; *Oh et al., 2012*; *Song et al., 2013*). Moreover, the temporal stability of the healthy human skin microbiome was recently investigated and diversity, skin site and individuality were all determinants of stability (*Flores et al., 2014*; *Oh et al., 2016*).

Despite the wealth of information regarding the skin microbiome of healthy humans, current knowledge in healthy domestic dogs (*Canis familiaris*) comes primarily from a study by *Rodrigues Hoffmann et al. (2014)*. This study showed that the canine bacterial community is diverse and quite variable across different body sites within the same dog, and across the same site in different dogs, suggesting that the skin microenvironment in dogs does not substantially impact the composition of its bacterial community. Similar to human skin, the most abundant phyla identified in dogs were Proteobacteria, Firmicutes, Actinobacteria, and Bacteroides. However, Actinobacteria has been shown to predominate in humans whereas it was less abundant in dogs (*Costello et al., 2009*; *Grice et al., 2008*, *2009*; *Grice & Segre, 2011*; *Rodrigues Hoffmann et al., 2014*). Host and environmental factors such as age, sex, breed, fleas, and time spent outside do not appear to influence the composition of the bacterial community in dogs. The study by *Rodrigues Hoffmann et al. (2014)* has improved our knowledge on the composition of the skin microbiome of healthy dogs, previously based only on culture-dependent methods. However, there is still much to be learned regarding the structure of the microbial communities that live on the skin of healthy dogs and the factors that shape these communities.

The primary aims of this study were to evaluate if there is a core bacterial community living on the skin of healthy domestic dogs from Minnesota, USA, and if body site, dog cohabitation and seasonality have an impact on this community.

## METHODS

### Study design

Healthy, privately owned paired dogs ($n = 40$) of various breeds from 20 households were enrolled in this study through the University of Minnesota Veterinary Medical Center. Dogs belonged to local clients or employees living in proximity to the Twin Cities, MN, USA. Owners signed an informed consent at the time of enrollment and were allowed to withdraw their dogs at any time during the study. To be included in the study the dogs were required to (1) be healthy based on a thorough history and clinical signs; (2) not receive any systemic or topical antimicrobial therapy for at least three months prior to enrollment; and (3) not be bathed for at least 30 days before inclusion.

Furthermore, in order for a household to participate in the study, none of the animals in the household could have skin or ear disease, and cohabiting dogs had to be living together for at least six months and spend at least 80% of the time together. The subjects consisted of 13 females and 27 males, with an average age of 7.6 years (Table 1). All animal work was carried out in accordance with the Institutional Animal Care and Use Committee at the University of Minnesota, protocol number 1108A03922.

At three timepoints spaced approximately three months apart (designated winter, spring, and summer), samples were collected from three sites on each dog (dorsal neck, axilla, and abdomen). Skin samples were collected by shaving a 10 cm² area at each site, and swabbing 25 times with a nylon-flocked swab (Copan Diagnostic Inc., Murrieta, CA, USA) moistened in SCF-1 (50 mM Tris, pH 7.6, 1 mM EDTA, 0.5% Tween-20). All samples were stored at 4 °C and processed within 2 h without freezing.

## Sample processing and sequencing

DNA was extracted using Mo Bio UltraClean®-htp 96 Well Microbial DNA kit (Mo Bio Laboratories, Carlsbad, CA, USA), according to the manufacturer's directions. Amplification of the 16S rRNA gene was performed using KAPA HiFidelity Hot Start Polymerase (Kapa Biosystems Inc., Wilmington, MA, USA) for two rounds of polymerase chain reaction (PCR) at the University of Minnesota Genomics Center (Minneapolis, MN, USA). For the first round, the V1V3F (5′-GTCTCGTGGGCTCGGAGATGTGTATAA GAGACAG**AGAGTTTGATCMTGGCTCAG**-3′) and V1V3R (5′-TCGTCGGCAGCGTCA GATGTGTATAAGAGACAG**ATTACCGCGGCTGCTGG**-3′) Nextera primers (Integrated DNA Technologies, Coralville, IA, USA) were used to amplify the V1–V3 hypervariable region using the following cycling parameters: one cycle of 95 °C for 5 min, followed by 20 cycles of 98 °C for 20 s, 55 °C for 15 s, and 72 °C for 1 min. The products were then diluted 1:100 and 5ul was used in a second round of PCR using forward (5′-**CAAGCA GAAGACGGCATACGA**GAT[i5]GTCTCGTGGGCTCGG-3′) and reverse (5′-**AATGATA CGGCGACCACCGA**GATCTACAC[i7]TCGTCGGCAGCGTC-3′) indexing primers (Integrated DNA Technologies, Coralville, IA, USA). The second PCR used the following cycling parameters: 1 cycle at 95 °C for 5 min, followed by 10 cycles of 98 °C for 20 s, 55 °C for 15 s, and 72 °C for 1 min. Pooled, size-selected samples were denatured with NaOH, diluted to 8 pM in Illumina's HT1 buffer, spiked with 15% PhiX, and heat denatured at 96 °C for 2 min immediately prior to loading. A MiSeq 600 (2 × 300 bp) cycle v3 kit (Illumina, San Diego, CA, USA) was used to sequence the samples.

## Data analyses

Following sequencing, sorting by barcode was performed to generate fastq files for each sample. Proximal and distal primers were trimmed from the sequence reads. Open referenced operational taxonomic unit (OTU) picking was used in QIIME (*Caporaso et al., 2010*) using uclust (*Edgar, 2010*). Potential chimeras were removed using ChimeraSlayer (*Haas et al., 2011*). OTUs present in negative control amplifications were also removed prior to subsequent analysis. After filtering due to low yield on some samples, a total of 40 abdomen, 46 dorsal neck, and 52 axilla samples (*n* = 138) were

**Table 1 Summary of enrolled dogs in this study.**

| Dog | Breed | Age | Gender | Coat |
| --- | --- | --- | --- | --- |
| 1A | Dachshund | 12 | FS | Long/medium |
| 1B | Dachshund | 7 | MN | Long/medium |
| 2A | Siberian Husky | 7 | MN | Long/medium |
| 2B | Mixed | 11 | FS | Long/medium |
| 3A | Labrador | 7 | MN | Long/medium |
| 3B | Yorkshire Terrier | 2 | MN | Long/medium |
| 4A | Chihuahua | 5 | MN | Short |
| 4B | Greyhound | 8 | FS | Short |
| 5A | Mixed | 11 | MN | Long/medium |
| 5B | Italian Greyhound | 9 | MN | Short |
| 6A | Boston Terrier | 12 | MN | Short |
| 6B | Boston Terrier | 13 | FS | Short |
| 7A | Newfoundland | 10 | MN | Long |
| 7B | Jack Russell Terrier | 3 | MN | Short |
| 8A | Dachshund | 3 | MN | Medium/wire |
| 8B | Dachshund | 3 | MN | Medium/wire |
| 9A | Mixed | 10 | MN | Long/medium |
| 9B | Greyhound | 9 | MN | Short |
| 10A | Miniature Poodle | 14 | MN | Short |
| 10B | Miniature Poodle | 6 | MN | Short |
| 11A | Malamute | 5 | FS | Long/medium |
| 11B | Siberian Husky | 6 | MN | Long/medium |
| 12A | Mixed | 4 | MN | Long/medium |
| 12B | Mixed | 8 | MN | Long/medium |
| 13A | Mixed | 10 | FS | Short |
| 13B | Mixed | 4 | FS | Short |
| 14A | Great Dane | 7 | FS | Short |
| 14B | Cavalier Spaniel | 2 | MN | Long/medium |
| 15A | Shih Tzu | 4 | FS | Long/medium |
| 15B | Shih Tzu | 4 | MN | Long/medium |
| 16A | Malamute | 6 | FS | Long/medium |
| 16B | Mixed | 4 | MN | Long/medium |
| 17A | Samoyed | 9 | MI | Long |
| 17B | Australian Shepherd | 14 | MN | Long/medium |
| 18A | Mixed | 10 | MN | Long/medium |
| 18B | Mixed | 10 | MN | Long/medium |
| 19A | Chihuahua | 5 | MN | Short |
| 19B | Chihuahua | 14 | FS | Short |
| 20A | Siberian Husky | 10 | FS | Long/medium |
| 20B | Siberian Husky | 6.5 | FI | Long/medium |

**Note:**
FS, female spayed; MN, male neutered; MI, male intact; FI, female intact.

retained for analysis following sequencing, quality filtering, and OTU clustering at 97% sequence similarity. Samples were rarefied to 5,000 high quality reads per sample.

QIIME was used for assessments of alpha diversity, beta diversity using Unifrac (*Lozupone & Knight, 2005*), phylogenetic classifications using the Greengenes database (*Cole et al., 2009*), and core bacterial community structure. Statistical analyses for differences in taxa between body site and season were performed using the Kruskal–Wallis test with correction for false discovery rate at 0.05. Statistical differences in overall community structure were performed in R using distance matrices analyzed via the ANOSIM command in QIIME (for beta diversity) and a nonparametric two sample *t*-test (for alpha diversity).

The raw data from this project is freely available at the Data Repository for the University of Minnesota (DRUM) at the following link: http://doi.org/10.13020/D6W01V.

## RESULTS

From 360 total samples, 138 samples were retained following removal of samples due to insufficient DNA for sequencing or low sequencing output (Table 2). Most failures were due to low DNA yield and the stringent conditions used for quality assessment and filtering of sequences. While all samples were subjected to DNA amplification and sequencing, many had fewer than 5,000 total reads which were subsequently discarded. Some of these samples were tested on subsequent runs with the same results. The total number of reads per sample in those used ranged from 5,016 to 297,512. Following filtering of OTUs not classified as bacteria, 6,966 OTUs were retained for downstream analyses. All samples were then rarefied to 5,000 sequences for subsequent analysis.

Samples were first taxonomically classified at the bacterial Class level by QIIME using OTUs and the Greengenes database (Fig. 1). When categorized by skin site or season, a range of differences was seen from sample-to-sample within each site, but these ranges did not visually differ between sites. The dominant classes were Actinobacteria (0–75.6%), Bacilli (0–62.2%), and Gammaproteobacteria (0–56.4%).

Operational taxonomic unit-based analyses were then performed using measures of alpha diversity and beta diversity. High Shannon diversity indices were observed across all samples whether grouped by season or skin site (Fig. 2). Using phylogenetic diversity across the entire tree and then Shannon diversity indices, no significant differences in alpha diversity were observed when grouping samples by age, sex, breed, hair type, season, or skin site.

Beta diversity was compared between samples using principal coordinate analysis plots (Fig. 3). There was no visual clustering of samples either by season or skin site. When assessing beta diversity using the ANOSIM function in QIIME, no significant differences were identified when grouping by age, sex, breed, hair type, or skin site. However, significant differences in community composition were identified when grouping by season ($P = 0.003$). When statistically different taxa at the OTU level were assessed by season, 13 total taxa were identified. Of these, the only differential taxa of appreciable relative abundance were those classified as Actinomycetales (class level) which was

**Table 2 Number of samples analyzed in this study by body site and season.**

|         | Abdomen | Axilla | Dorsal neck | Total |
|---------|---------|--------|-------------|-------|
| Spring  | 15      | 19     | 26          | 60    |
| Summer  | 16      | 14     | 7           | 37    |
| Winter  | 9       | 19     | 13          | 41    |
| Total   | 40      | 52     | 46          | 138   |

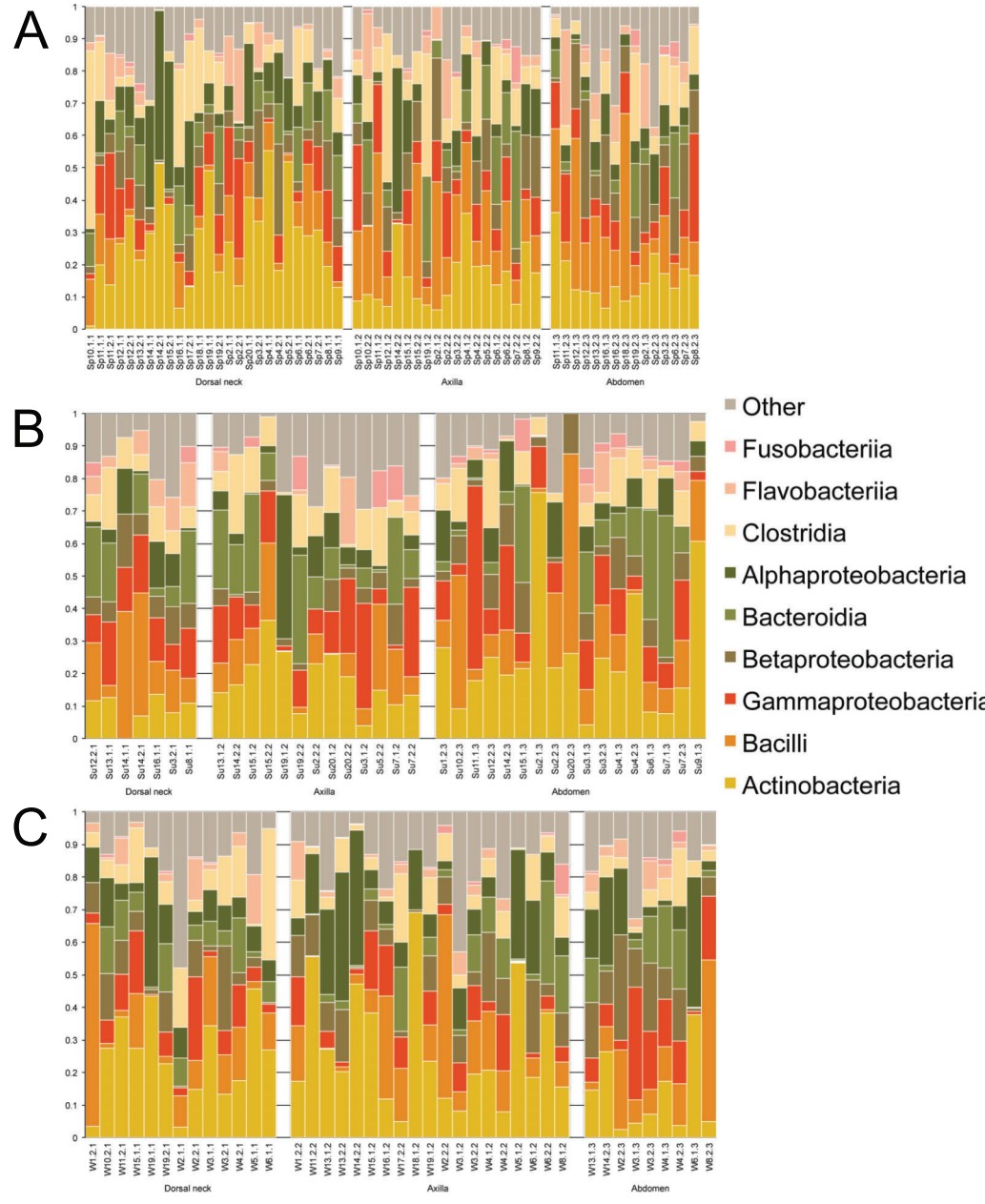

**Figure 1 Individual samples classified by Greengenes according to bacterial class, grouped by skin site, and ordered by sample number.** (A) Spring samples, (B) Summer samples, and (C) Winter samples.

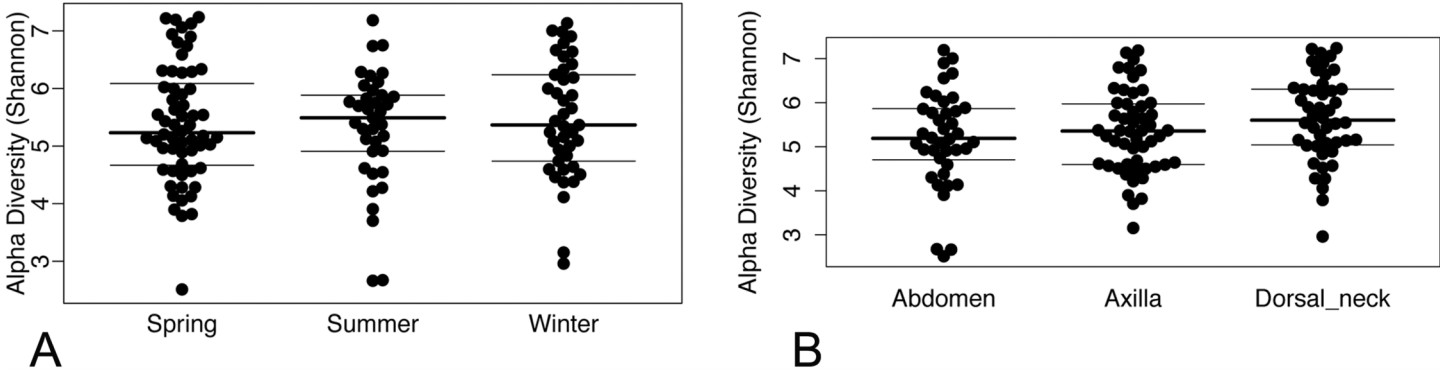

**Figure 2** (A) Beeswarm plots of Shannon diversity values with median and interquartile ranges for samples grouped by season and (B) skin site.

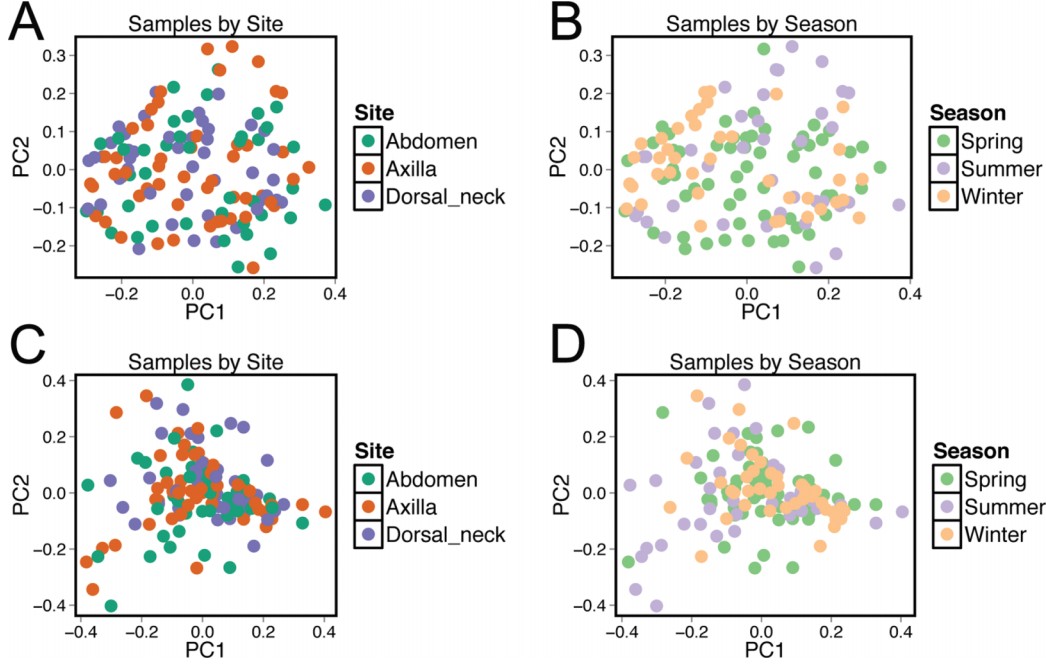

**Figure 3** **Principle coordinate analysis (PCoA) plots of individual samples.** Samples are colored by skin site using unweighted matrices (A) and weighted matrices (C), or colored by season site using unweighted matrices (B) and weighted matrices (D).

of lower relative abundance in the summer, and *Actinomyces* (genus) which was also of lower relative abundance in the summer.

Unifrac distances were used to examine the dissimilarity between samples grouped by a variety of criteria, including sex (male or female), hair type (short or long), breed, age, skin site, season, household, and individual dog (Fig. 4). In the analyses, 'same' indicates average dissimilarity between samples within the same grouping, whereas 'different' indicates average dissimilarity between samples of different groupings. When comparing 'same' samples to 'different' samples for each category using unweighted Unifrac distances, significant differences in distance were identified when

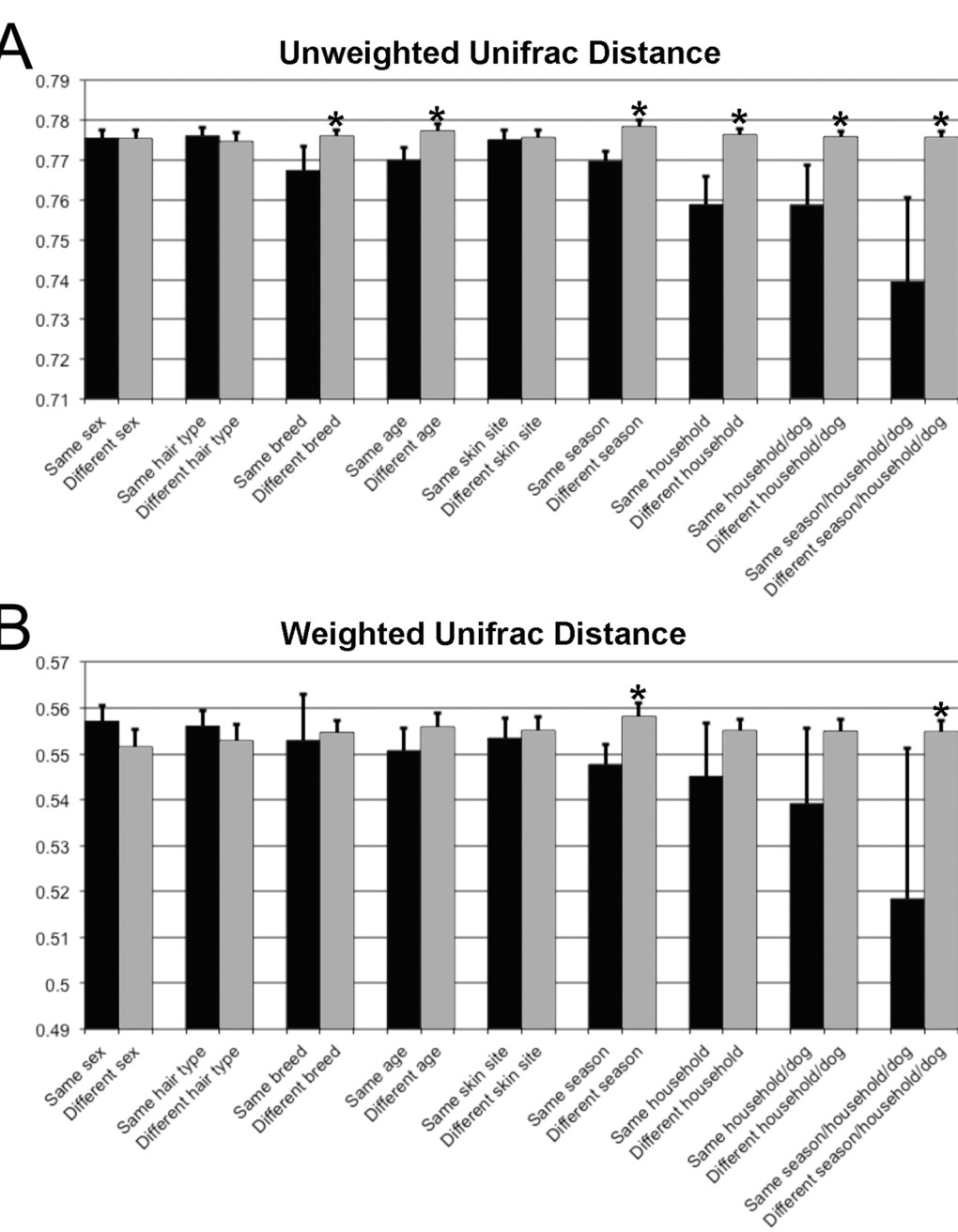

**Figure 4 Comparison of unweighted and weighted Unifrac distance matrices between samples when grouped by a variety of different criteria.** 'Same' indicates average dissimilarity between samples within the same grouping, whereas 'different' indicates average dissimilarity between samples between different groupings. (A) Uses unweighted Unifrac distances and (B) uses weighted Unifrac distances. Those comparisons that are statistically significant ($P < 0.05$) are indicated with asterisks. Error bars represent 95% confidence intervals.

grouped by season ($P = 3.5 \times 10^{-8}$), household ($P = 1.2 \times 10^{-7}$), breed ($P = 0.003$), and age ($P = 1.7 \times 10^{-5}$). The largest differences between 'same' and 'different' samples were observed when categorizing by same dogs within the same season (across multiple skin sites) (Fig. 4).

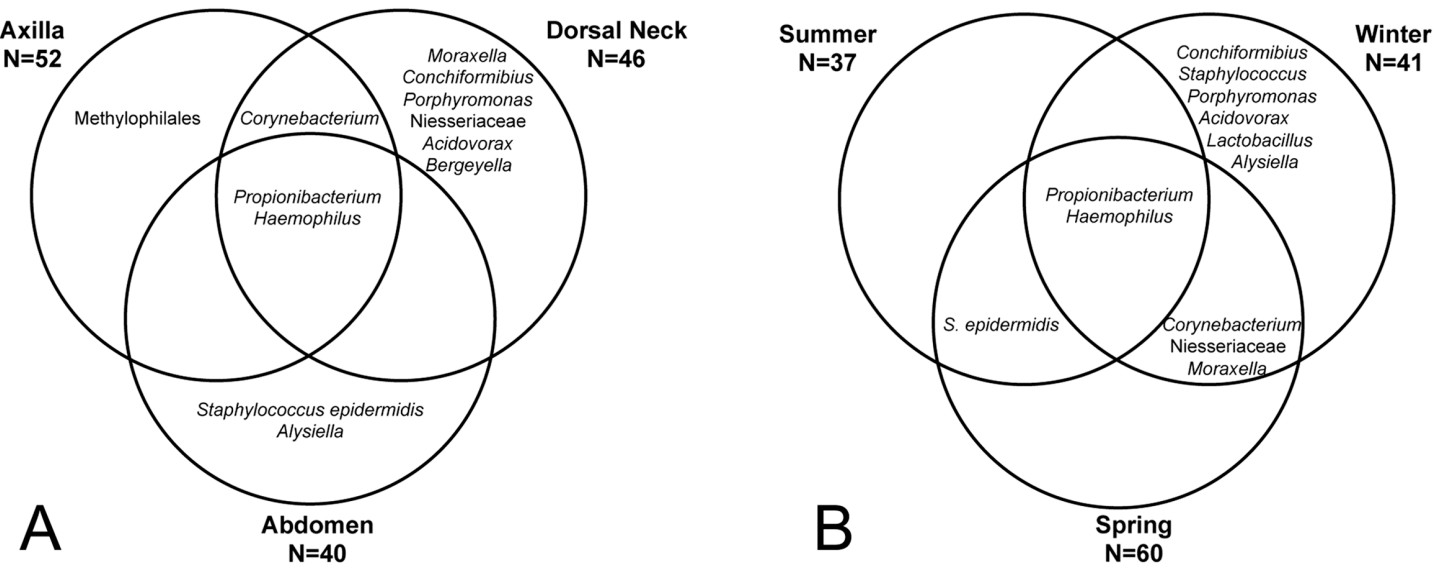

**Figure 5 OTUs present in >50% of all samples by group.** (A) Depicts samples by skin site, and (B) depicts samples by season.

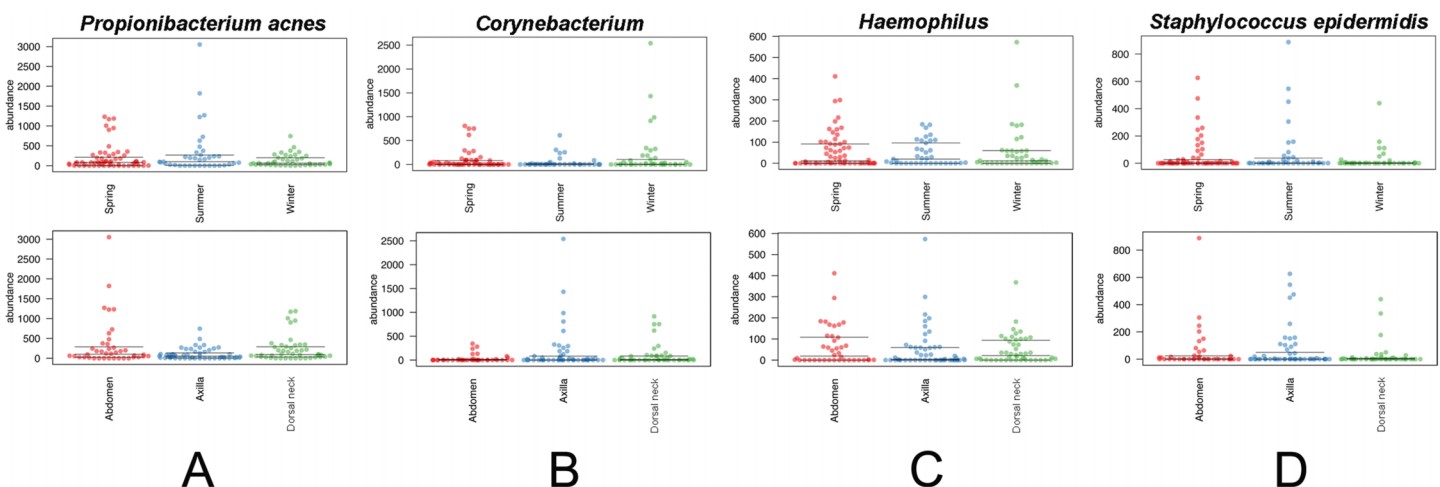

**Figure 6 Distribution of selected OTUs identified and classified using the Greengenes database.** Beeswarm plots depict individual sample OTU abundance based on 5,000 normalized reads per sample. Boxplots indicate median and quartile ranges for each OTU. The top plot for each OTU categorizes samples by season, whereas the bottom plots categorize samples by skin site. (A) *Propionibacterium acnes*, (B) *Corynebacterium*, (C) *Haemophilus*, and (D) *Staphylococcus epidermidis*.

The core and dominant OTUs in canine skin were assessed based upon their prevalence amongst samples and their relative abundance. A cutoff of 50% prevalence across samples was used to assess the core bacterial community because of the lack of any OTUs present in 80% or greater samples in each group, the established standard for core microbiome definition (*Li, Bihan & Methé, 2013*) (Fig. 5). Only two OTUs were identified as core to all groups by season and by skin site. An OTU classified as *Propionibacterium acnes* was present in >80, >75, and >90% of spring, summer, and winter samples, respectively, and in >80, >80, and >90% of abdomen, axilla, and dorsal neck samples,

respectively. A second OTU classified as *Haemophilus* was present in >60, >60, and >65% of spring, summer, and winter samples, respectively, and in >60, >55, and >70% of abdomen, axilla, and dorsal neck samples, respectively. An OTU classified as *Corynebacterium* was dominant in relative abundance across many samples, but highly variable (Fig. 6). In particular, it was more prevalent across samples in winter (>95%) and spring (>80%), and samples from the axilla (>80%) and dorsal neck (>90%). An OTU classified as *Staphylococcus epidermidis* was more prevalent amongst abdomen samples (>55%), and samples from spring (>50%) and summer (>55%).

## DISCUSSION

A wealth of data exists for the bacterial communities inhabiting human skin, but less in known about their counterpart domestic dogs. The purpose of this study was to examine the skin bacterial communities of domestic dogs to assess the effects of cohabitation and season, and to determine if a core skin bacterial community could be identified across a diverse group of animals. The results of these analyses suggest that the canine skin bacterial community is highly diverse and highly variable. *Rodrigues Hoffmann et al. (2014)* came to the same conclusion when examining 12 skin sites from 12 healthy dogs. They found that *Ralstonia* was the most abundant genus identified across skin samples, followed by *Moraxella* and *Porphyromonas*. In contrast, we identified *P. acnes* as the most abundant OTU, followed by *Corynebacterium* and *Porphyromonas*. Interestingly, a study using culture-based methods found *P. acnes* in the epidermis and hair follicles of seven of 11 (63.6%) dogs suggesting that this bacterium is indeed an important skin resident of dogs (*Harvey, Noble & Lloyd, 1993*). The differences between this and Hoffman's study are likely a factor of the variability of the canine skin microbiota, since each dominant OTU identified here was indeed highly variable across samples, and/or DNA extraction techniques, primer selection, and PCR parameters. Thus, there is certainly a distinct collection of bacterial species that inhabit the skin of dogs that differs from that of humans, but it is likely impacted dramatically by the innate behaviors of the dog compared to humans.

There were no significant differences in overall bacterial community structure between the three skin sites examined in this study, but there was a significant effect when samples were grouped by timepoint. Again, while statistically significant, the variability between samples of the same timepoint (season) dampened the effect. Actinobacteria appeared to be found at lower relative abundance in the summer samplings as compared to the winter and spring samplings. It is unclear if this is a meaningful effect, though, as intuitively one would expect Actinobacteria to be at higher abundance in the summer when dogs are spending more time outdoors since Actinobacteria are ubiquitous in soil and water. It should also be cautioned that only one timepoint per season was assessed. Sampling across multiple years would be required to make definitive statements regarding a true seasonal effect versus a sampling effect.

Unifrac distances revealed that there is a significant cohabitation effect on the dog skin bacterial community. That is, dogs that live together have significantly more similar bacterial communities than dogs not living together. Furthermore, samples from the same

dog within a household (at different skin sites) amplify this effect. This supports the conclusions that (1) there is significant sharing of bacteria between dogs within the same household, and (2) skin bacterial communities within the same dog across body sites are more similar than non-self samples. Thus, the individual dog appears to have its own unique bacterial community that is consistent across multiple skin sites within the animal. Notably, the effect of cohabitation on the dog skin bacterial communities observed here was less than the increased sharing of microbes between household human partners mediated by household dogs, observed by *Song et al. (2013)*. This is expected, though, as the referenced study examined owner hands. Certainly, most dogs within the same household are less likely to have direct intimate contact with each other as compared to the owner–dog interaction.

Finally, there has been much work aimed at the effects of dog ownership on allergies and asthma in humans, exemplified by a recent study demonstrating that exposure to pets and farm animals reduces the risk of childhood asthma (*Fall et al., 2015*). Our data and the results of others indicate that dogs provide a rich source of environmental bacteria to the household, and a study using vacuum settled dust found that dog ownership has also been shown to positively impact the diversity and evenness of bacterial communities in the home (*Kettleson et al., 2015*). This further indicates the role of the household dog in facilitating the introduction and dissemination of a rich bacterial community throughout the household.

## ACKNOWLEDGEMENTS

Bioinformatics were supported using tools available from the Minnesota Supercomputing Institute.

### Funding

This project was supported by grant D13CA-037 from the Morris Animal Foundation awarded to Timothy J. Johnson and Sheila Torres, and the National Institutes of Health PharmacoNeuroImmunology Fellowship (NIH/NIDA T32 DA007097-32) awarded to Jonathan B. Clayton. The funders had no role in study design, data collection and analysis, decision to publish, or preparation of the manuscript.

### Grant Disclosures

The following grant information was disclosed by the authors:
Morris Animal Foundation: D13CA-037.
National Institutes of Health PharmacoNeuroImmunology Fellowship: NIH/NIDA T32 DA007097-32.

### Competing Interests

The authors declare that they have no competing interests.

## Author Contributions

- Sheila Torres conceived and designed the experiments, performed the experiments, contributed reagents/materials/analysis tools, wrote the paper, prepared figures and/or tables, reviewed drafts of the paper.
- Jonathan B. Clayton analyzed the data, reviewed drafts of the paper.
- Jessica L. Danzeisen performed the experiments, reviewed drafts of the paper.
- Tonya Ward analyzed the data, contributed reagents/materials/analysis tools, reviewed drafts of the paper.
- Hu Huang analyzed the data, contributed reagents/materials/analysis tools, reviewed drafts of the paper.
- Dan Knights: analyzed the data, contributed reagents/materials/analysis tools, reviewed drafts of the paper.
- Timothy J. Johnson conceived and designed the experiments, performed the experiments, analyzed the data, contributed reagents/materials/analysis tools, wrote the paper, prepared figures and/or tables, reviewed drafts of the paper.

## Animal Ethics

The following information was supplied relating to ethical approvals (i.e., approving body and any reference numbers):

All animal work was carried out in accordance with the Institutional Animal Care and Use Committee at the University of Minnesota, protocol number 1108A03922.

## Data Deposition

The raw data from this project is freely available at the Data Repository for the University of Minnesota (DRUM) at the following links: http://hdl.handle.net/11299/183048 or http://doi.org/10.13020/D6W01V.

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
