# Peer review of "Diverse bacterial communities exist on canine skin and are impacted by cohabitation and time"

_PeerJ, doi:10.7717/peerj.3075_

## Round 0.1 · original submission · Minor Revisions

· Academic Editor

Minor Revisions

Overall a well done study. Please address the reviewers comments as well as these 3 comments. The revisions are all relatively minor.
Define recent advances (line 60).
Line 70 - were these 2 studies on healthy subjects?
Line 114 - how many minutes? List this instead of as soon as possible.

·

Basic reporting

I find the article by Torres et al. to have been well written, using proper English throughout. As far as I could tell, the introductory material provided in the “Introduction” was very informative and thorough. I did have one very minimal concern related to the very beginning of the “Introduction”. From line 58 to 66 it was not clear that the “bacterial communities” they were talking about were found on human or dog skin. I think they need to be sure not to confuse just which “bacterial communities” they are most interested in, and that the rest of the article will focus on. The references they’ve cited in the “Introduction” appear to be timely and cover a range of similar studies.

The overall structure of the article appears to be pretty normal. As far as I can tell the authors have conformed to the PeerJ Standards.

All figures that I saw appeared to be well composed. They were all introduced and discussed in the text. I found that the figures helped me better understand some of the more complex aspects of genomics, especially with regard to statistical treatment of this data.

One bit of data that intrigued me, but was not addressed by the authors, was the fact that obligate anaerobes (Clostridium sp.) would be found living on the skin of dogs. Presumably, detection of these bacteria on the skin of the dogs would require that spores associated with the bacteria be present, since the skin of these dogs is assumed to be fully aerobic in nature. Is there any link between the dogs time spent outdoors, where they might get soil or mud (which might contain the spores) on their skin, and the presence of these obligate anaerobes on their skin? This would be an interesting line of discussion, especially if dogs might be considered to be a vector for environmental bacteria to the inside of human domiciles (a point the authors make in their conclusions). Bringing spores for C. botulinum or C. tetani from soil outdoors into the living space for humans might not be considered a good thing!

Lastly, raw data was supplied with the manuscript. As far as I know everything I needed to consider the manuscript was provided.

Experimental design

The article conforms nicely with the scope of PeerJ. The authors have clearly stated their objective in conducting this research (lines 89 to 91). As noted in my consideration of the Validity of the Findings, I do think that the authors should indicate that the dogs (Canis familiaris) they are studying are a subset of the genus Canis and do not represent the entire genus. I do think that a study of wild dogs might provide a better control for a study seeking to determine “core bacterial community” structure of animals in this genus.

One other question I have about the study is the degree to which the human hosts had interacted with the subject dogs prior to the study. As noted above, with the dogs most likely all coming from Minnesota, and with the cold winters there, these dogs might be expected to have more direct interaction with their hosts indoors during the winter. How would the microbial communities of dogs that tend to live outdoors all year (say, living in a dog house) in warmer climates differ?

From what I’ve read, and if my lab were outfitted to allow these studies, I do think that adequate description of the methods used was provided in the manuscript. The lab analyses preformed seemed to be rigorous.

Overall, if the authors were to indicate that they are simply focused on domesticated dogs, their study methods are well presented.

Validity of the findings

The authors of this study sought to determine if a “core bacterial community” could be found for dogs. They considered whether different sites on the dogs, dog cohabitation, and season affected the communities of bacteria they detected. Based on a quick search of the literature there are few published studies that have done this.

I do have some questions, specifically whether the aims of the authors were to determine a “core bacterial community” for dogs as a species, or domesticated dogs in Minnesota? There was no indication that any of the dog pairs lived anywhere but in Minnesota. The seasonal affects noted might be different if dogs from households in Miami were added to the study. Also, since prior published studies have indicated that both human and dog skin microbiomes tend to be affected by human – dog interactions, I find it unusual that the authors would seek to find a “core bacterial community” on dogs that had lived an average of 7.6 years with human hosts. Wouldn’t it be better to have some sort of control group of coyotes, or even wolves to see if there were a “core bacterial community” associated with Canines, rather than just for domesticated dogs? If the authors cannot collect samples from “wild” dogs, then they should modify the scope of their study to that of domesticated dogs.

As far as I could tell the data seemed to be robust. I am not very familiar with some of the analytical tests used with genomic data, so I assume they have been used properly. Having a total of 40 dogs in 20 pairs should be adequate, but, I don’t really see any sort of control in the dogs used. I think that the study would be really unique if wild dogs (or other Canines) had been used as controls.

Additional comments

After reading this manuscript I feel that I have learned a good deal about microbial communities on domestic dogs. I think that if a few additional controls were used the study would be excellent.

·

Basic reporting

Clearly written manuscript describing and analyzing the characteristics of the skin microbiota of the dog. 40 dogs belonging to 20 households are sampled three times (Spring, Summer and Winter). The manuscript contains adequate background, M&M information, figures and tables. Raw data are accesible at the Data Repository of the Un. Minnesota. Although the conclusions are not striking the study is interesting because of the scarce data about the skin microbiota of the dog.

Experimental design

The experimental design is correct and the methods are clearly described. Main concern is the loss of samples. From the 360 samples expected, only 138 were finally analyzed. According to the authors these samples contained insufficient DNA or they had low sequencing output. The authors should explain:

(1) What caused this situation? Inadequate sampling? Inadequate post - sampling preservation of the samples?

(2) Did this issue impact the quality of the statistical analysis? For instance, there were enough samples from all haircoats/sites (there were only 7 samples from/dorsal neck in the summer time point).

Validity of the findings

In general the authors are really prudent stating conclusions. Most of the paper is descriptive. The few conclusions (significant differences in bacterial community structure between samples when samples were stratified by season, but not by dog sex, age, hair type or skin site) seem to be sound. The authors however should be careful attributing te changes to the "season". The way to be sure that the changes were linked to the season ("summer") would be collecting samples in the same season for two different years (summer 2014 and summer 2015) and showing that the were similar (or more similar the other samples). The observed difference between samples can be consequence of the time progression more than to the season.

It is less clear (not enough robust) the conclusion regarding the core microbiome. The authors propose to define a microorganism as part of the core microbiome when is present in over 50% of the samples. This is a weird definition of the "core"; can the authors please give preference for this criterium? Most authors define the "core" based on prevalence in the samples (ubiquity; usually over 80%) and abundance (see for instance: Analyses of the Stability and Core Taxonomic Memberships of the Human Microbiome. Kelvin Li, Monika Bihan, Barbara A. Methé. In PLoS ONE 2013). The authors should clarify their definition of "core". The variability in the samples is so high that in my opinion is not possible to define the core microbiome of the canine skin based only on these results.

Additional comments

---

## Round 0.2 · accepted · Accept

· Academic Editor

Accept

Thank you for making the revisions - the rebuttal was clear and helpful.